# State-of-the-Art Techniques for Diagnosis of Medical Parasites and Arthropods

**DOI:** 10.3390/diagnostics11091545

**Published:** 2021-08-26

**Authors:** Pichet Ruenchit

**Affiliations:** Department of Parasitology, Faculty of Medicine Siriraj Hospital, Mahidol University, Bangkok 10700, Thailand; pichet.rue@mahidol.edu; Tel.: +66-2-419-6484

**Keywords:** arthropod, artificial intelligence, diagnosis, DNA barcoding, geometric morphometrics, parasite, parasitic disease, parasitology

## Abstract

Conventional methods such as microscopy have been used to diagnose parasitic diseases and medical conditions related to arthropods for many years. Some techniques are considered gold standard methods. However, their limited sensitivity, specificity, and accuracy, and the need for costly reagents and high-skilled technicians are critical problems. New tools are therefore continually being developed to reduce pitfalls. Recently, three state-of-the-art techniques have emerged: DNA barcoding, geometric morphometrics, and artificial intelligence. Here, data related to the three approaches are reviewed. DNA barcoding involves an analysis of a barcode sequence. It was used to diagnose medical parasites and arthropods with 95.0% accuracy. However, this technique still requires costly reagents and equipment. Geometric morphometric analysis is the statistical analysis of the patterns of shape change of an anatomical structure. Its accuracy is approximately 94.0–100.0%, and unlike DNA barcoding, costly reagents and equipment are not required. Artificial intelligence technology involves the analysis of pictures using well-trained algorithms. It showed 98.8–99.0% precision. All three approaches use computer programs instead of human interpretation. They also have the potential to be high-throughput technologies since many samples can be analyzed at once. However, the limitation of using these techniques in real settings is species coverage.

## 1. Introduction

A parasitic disease is an infectious disease caused by a parasite, an organism that lives in or on a host organism. After infection, the parasite induces pathogenesis at the different organ systems that it invaded by mechanical damage, chemical damage, organ obstruction, hyperplasia, allergy to toxic wastes, or indirect effects, such as transmission of other microbes [1]. The parasites causing these pathogeneses in a human host are known as “medical parasites”. They include Protista (*Acanthamoeba* spp. (Volkonsky) (Longamoebia: Acanthamoebidae), *Blastocystis* spp. (Alexieff) (Opalinata: Blastocystidae), *Entamoeba* spp. (Casagrandi and Barbagallo) (Mastigamoebida: Entamoebidae), *Giardia lamblia* (Lambl) (Diplomonadida: Hexamitidae), *Naegleria fowleri* (Carter) (Schizopyrenida: Vahlkampfiidae), and *Plasmodium* spp. (Marchiafava and Celli) (Haemosporida: Plasmodiidae)); Trematoda (*Opisthorchis viverrini* (Poirier) (Opisthorchiida: Opisthorchiidae) and *Paragonimus westermani* (Kerbert) (Plagiorchiida: Troglotrematidae)); Cestoda (*Hymenolepis nana* (Bilharz) (Cyclophyllidea: Hymenolepididae) and *Taenia saginata* (Goeze) (Cyclophyllidea: Taeniidae)); and Nematoda (*Ascaris lumbricoides* (Linnaeus) (Rhabditida: Ascarididae), *Enterobius vermicularis* (Linnaeus) (Rhabditida: Oxyuridae), *Strongyloides stercoralis* (Bavay) (Rhabditida: Strongyloididae), *Trichinella spiralis* (Owen) (Trichinellida: Trichinellidae), and *Wuchereria bancrofti* (Cobbold) (Rhabditida: Onchocercidae)) [1]. Nowadays, parasitic diseases are responsible for significant morbidity and mortality in both tropical and subtropical regions [2]. Globally, there are approximately 3.5 billion cases of intestinal parasitic infections [3]. In Thailand, the prevalence of parasitic infections has steadily decreased due to improved sanitation and personal hygiene. However, sporadic cases of parasitic infections have been reported in different geographical areas of Thailand [4,5], especially in crowded settings or among immunocompromised individuals, such as people living with human immunodeficiency virus (HIV), cancer patients, and immunosuppressed patients.

Many arthropods also play a critical role in human health, such as Insecta (*Anopheles* spp. (Meigen) (Diptera: Culicidae), *Cimex* spp. (Linnaeus) (Rhynchota: Cimicidae)*, Paederus fuscipes* (Fabricius) (Coleoptera: Staphylinidae), *Pediculus humanus* (Linnaeus) (Phthiraptera: Pediculidae), and *Phthirus pubis* (Linnaeus) (Phthiraptera: Pthiridae)) and Arachnida (*Sarcoptes scabiei* (De Geer) (Sarcoptiformes: Sarcoptidae) and *Dermatophagoides* spp. (Bogdanov) (Sarcoptiformes: Pyroglyphidae)). These are generally known as “medical arthropods”, and they cause both direct and indirect effects in humans globally. For example, *P. humanus*, *P. pubis*, and *S. scabiei* exert direct effects on humans by causing ectoparasitosis (pediculosis, phthiriasis, and scabies). *Chrysomya megacephala* (Fabricius) (Diptera: Calliphoridae), *Cochliomyia hominivorax* (Coquerel) (Diptera: Calliphoridae), and *Wohlfahrtia magnifica* (Schiner) (Diptera: Sarcophagidae) cause cutaneous myiasis (wound myiasis), while *Cordylobia anthropophaga* (Blanchard) (Diptera: Calliphoridae) and *Dermatobia hominis* (Linnaeus) (Diptera: Oestridae) cause furuncular myiasis in humans [6]. Moreover, many groups of arthropods are causative agents of envenomization, allergic reaction, annoyance, and delusory parasitosis. They also serve as vectors and intermediate hosts of human pathogens, such as fleas, hematophagous flies, mosquitoes, and ticks [6]. Some arthropods display the potential to cause outbreaks in overcrowded areas, such as scabies and pediculosis outbreaks in military camps and schools, respectively [7,8].

Laboratory investigations to identify the causative organisms of parasitic and arthropod diseases play a crucial role in differential diagnosis, definitive diagnosis, medical treatment, and outbreak control. The investigations also serve as a surveillance tool to improve human health status. There are various kinds of laboratory techniques to detect medical parasites and arthropods. However, the diagnosis of parasitic and arthropod diseases is still largely based on traditional methods, such as gross and microscopic examinations, particularly in resource-poor clinical settings. There is wide concern about the limitations of these conventional methods, such as that they are time consuming and labor intensive, have low sensitivity and accuracy, and need skilled technicians. New technologies are therefore gaining more research and development attention in order to overcome those limitations, especially to avoid interpretation bias from physicians, by using computer programs instead. Recently, advanced diagnostic approaches for parasite and arthropod identifications have emerged to avoid the pitfalls, such as DNA barcoding techniques, geometric morphometric analysis, and artificial intelligence technology. These techniques utilize computer programs instead of humans for data analysis and interpretation. Details of these emerging technologies are extensively described in this review article.

## 2. Conventional Diagnostic Methods for Medical Parasite and Arthropod

Before getting into the details of the new advanced technologies for diagnosing medical parasites and arthropods, the conventional diagnostic methods are firstly briefly reviewed. Their sensitivity, specificity, accuracy, advantages, and disadvantages are also discussed to understand the purpose of the development of the novel techniques.

### 2.1. Gross Examination

Generally, gross examination of feces and parasites is performed in routine laboratory diagnoses of parasitic diseases (Table 1). Fecal specimens are grossly examined for color, odor, and consistency and the presence of blood and mucus [9]. These characteristics can be used to diagnose the types of the suspected organisms. Whole, portions of, and segments of parasites or larvae are also examined by gross parasite examination to diagnose parasitic diseases like taeniasis and paragonimiasis [10]. The adult and larval stages of arthropods that caused myiasis are also diagnosed by this method using taxonomic keys (Table 1) [11].

### 2.2. Microscopic Examination

Microscopic examination is a basic and gold standard method for the diagnosis of parasitic infections. It has been used for parasitological diagnosis for several hundred years (Figure 1) [12]. Direct wet smear, concentration, staining, and scotch tape techniques are used to detect ova and parasite (Table 1) [13,14,15,16,17]. For malaria diagnoses, microscopic examination of Giemsa-stained thick and thin blood films is a standard practice, even though several approaches have emerged, such as dipstick antigen detection and polymerase chain reaction (PCR). This is due to the very low cost of microscopic examination compared to PCR and antigen detection, meeting the high demand for tests in endemic areas with low resources. In the case of arthropod identification, the skin-scraping technique is used for scabies diagnosis, while microscopy is used for pediculosis and phthiriasis diagnoses [18,19]. However, these methods are labor-intensive and time-consuming [12]. Furthermore, the sensitivity and specificity are still low [20].

**Table 1 diagnostics-11-01545-t001:** Conventional methods for diagnosis of medical parasites and arthropods.

Types of Examination	Method Names	Diagnostic Tool of		References
		Parasite ^1^	Arthropod ^2^	
Gross	Gross examination	Yes	Yes	[10,11]
Microscopic	Direct wet smear	Yes	No	[14]
	Concentration technique	Yes	No	[15]
	Staining technique	Yes	No	[16]
	Scotch tape technique	Yes	No	[17]
	Skin scraping technique	No	Yes	[18]
Culture	Harada–Mori technique	Yes	No	[21]
	Baermann technique	Yes	No	[22]
	Charcoal culture technique	Yes	No	[23]
	Agar plate culture technique	Yes	No	[24]
Immunological	Enzyme-linked immunosorbent assay (ELISA)	Yes	No	[25]
	Immunoblot assay	Yes	No	[25]
Molecular biology	Polymerase chain reaction (PCR)	Yes	Yes	[12,26,27]
	Loop-mediated isothermal amplification (LAMP)	Yes	Yes	[20,26,28]
	xMAP assay (Luminex)	Yes	No	[26,29]
	Restriction fragment length polymorphism (RFLP)	Yes	Yes	[26,30]
	Next-generation sequencing (NGS)	Yes	Yes	[20,31]
	Mass spectrometry (MS)	Yes	Yes	[12,32]
	Biosensors	Yes	No	[20,33]

^1^ Diagnostic tool that has been reported to be used to identify parasites. ^2^ Diagnostic tool that has been reported to be used to identify arthropods.

### 2.3. Culture Technique

Fecal culture methods are utilized for the larval-stage detection of nematodes, such as *S. stercoralis* and hookworms [13]. The methods include Harada–Mori, Baermann, charcoal, and agar plate culture techniques (Table 1) [21,22,23,24]. Most are used as diagnostic tools for *S. stercoralis* [21,22,23,24]. In addition, culture techniques are used for the diagnosis of many medical protozoa, such as *Acanthamoeba* spp., *E. histolytica*, *Leishmania* spp. (Borovsky) (Trypanosomatida: Trypanosomatidae), *N. fowleri,* and *Trichomonas vaginalis* (Donné) (Trichomonadida: Trichomonadidae) [34]. The cultivation of these parasites requires special media, such as Peptone-Yeast Extract-Glucose (PYG) and Nelson’s media for *Acanthamoeba* spp. and *N. fowleri*, respectively. Therefore, their use tends to be restricted to certain laboratories, particularly clinical research centers.

### 2.4. Immunological Examination

To increase the sensitivity and specificity of parasite detection, immunologically based methods were developed. They are tools for diagnosing medical parasites based on antigen or antibody detections. Enzyme-linked immunosorbent assay (ELISA) is a well-known immunological method that can be used as a gold standard diagnostic (Table 1) [25]. It was used to detect *Angiostrongylus cantonensis* (Chen) (Strongylida: Angiostrongylidae)*, E. histolytica*, and *Taenia* spp. infections. Immunoblot assay is another technique that was used to diagnose gnathostomiasis, schistosomiasis, strongylodiasis, and hydatidosis [25]. Nonetheless, these techniques have limitations: they have high cross-reactivity, and they are time consuming, labor intensive, and unable to detect low-level antibodies in patients with clinical onset. Importantly, costly reagents are required.

### 2.5. Molecular Biology Examination

One of the most promising molecular techniques for the diagnosis of medical parasites and arthropods is the polymerase chain reaction (PCR) [12]. It was used to detect non-intestinal protozoa, such as *Plasmodium* spp. and *Leishmania* spp. [26]. Moreover, it has been utilized to detect *S. scabiei* from the skin scrapings of suspected scabies patients [27]. Compared with microscopic and immunological examinations, PCR displays more advantages in terms of sensitivity and specificity, and multiplexed PCR can detect several parasite-specific sequences in the same reaction. However, inhibitors from stool samples and cross-contamination problems still have the most impact on the sensitivity and specificity of this method. Moreover, other techniques can be applied for parasite and arthropod identifications, such as loop-mediated isothermal amplification (LAMP), xMAP assay, restriction fragment length polymorphism (RFLP), next-generation sequencing (NGS), mass spectrometry (MS), and biosensors (Table 1) [12,20,26,28,29,30,31,32,33].

As shown in Table 1, several molecular techniques are promising tools for the diagnosis of medical parasites and arthropods. This is because of their high sensitivities, high specificities, short processing times, relatively low human-resource requirements, and ability to be used for mass screening and point-of-care testing. However, most require costly reagents and equipment as well as skilled technicians to interpret their results [12]. To conquer these limitations, advanced diagnostic tools have recently been developed for parasite and arthropod identifications. These include the DNA barcoding technique, geometric morphometric analysis, and artificial intelligence technology. To avoid the risk of misdiagnosis, all utilize computer programs for data analysis and interpretation instead of human diagnosticians. Moreover, most do not require expensive reagents or equipment other than computers.

## 3. Advanced Approaches for Medical Parasite and Arthropod Diagnoses

In this review, the principles, steps, and applications to both the parasite and arthropod diagnoses of the DNA barcoding technique, geometric morphometric analysis, and artificial intelligence technology are detailed. Their sensitivities, specificities, accuracies, advantages, and disadvantages are also discussed and compared with those of the conventional methods.

### 3.1. DNA Barcoding Technique

It is widely recognized that skilled technicians are needed for the interpretation of the most traditional parasitological methods, including molecular techniques [12]. The identification of parasites and arthropods is fairly complicated since they show high diversity in terms of the quantity and variety of their stage-specific morphologies [35,36]. Hence, unpredictable wrong diagnoses normally occur during laboratory investigations. This issue affects the therapeutic intervention efficacy of parasitic diseases. Therefore, new approaches, such as the DNA barcoding technique, were developed and utilized for parasite and arthropod identifications during the last decade [37].

The concept of DNA barcoding was first proposed by Hebert and colleagues in 2003 (Figure 1) [37]. The principle of this technique is the analysis of the nucleotide sequence of a short DNA fragment (400–800 bases) called “DNA barcode”. A partial fragment of mitochondrial cytochrome *c* oxidase subunit I (COI) or internal transcriber spacer (ITS) is universally used as the standardized barcode region [38,39]. The reason for choosing the mitochondrial element as a target of sequence analysis instead of the nuclear gene is its lack of introns and its limited exposure to recombination [40]. The steps of parasite and arthropod diagnoses using the DNA barcoding technique are illustrated in Figure 2. DNA is extracted from clinical specimens of feces, urine, blood, sputum, aspirate, corneal swab, biopsy, or a gross sample. The DNA barcode is amplified by PCR using the universal primers specific to the conserved flanking regions of the barcode sequence. These primers are specific to the COI gene of most organisms; therefore, almost all groups of parasites and arthropods can be diagnosed [41]. The PCR amplicon is then submitted for nucleotide sequencing and subsequently analyzed for homology to other sequences recorded in the internationally reference databases, such as Barcode of Life Data (BOLD) system and GenBank. Based on this technique, the unknown samples of parasites and arthropods can be automatically identified at both the genus and species levels by efficient algorithms without any bias from humans.

The DNA barcoding technique showed high accuracy in parasite and arthropod diagnoses (Table 2). Ondrejicka et al. (2014) reported that the technique had a 95% accuracy in parasitological diagnosis. This is due to the high specificity of the nucleotide sequence of each organism. Nowadays, the DNA barcode database encompasses more than 1400 parasite and arthropod species. They include *A. lumbricoides*, *Ancylostoma duodenale* (Dubini) (Strongylida: Ancylostomatidae)*, Clonorchis sinensis* (Cobbold) (Opisthorchiida: Opisthorchiidae), *Fasciola hepatica* (Linnaeus) (Plagiorchiida: Fasciolidae), *Necator americanus* (Stiles) (Strongylida: Ancylostomatidae), *Onchocerca* spp. (Bickel) (Rhabditida: Onchocercidae), *Paragonimus* spp., *Schistosoma* spp. (Weinland) (Strigeidida: Schistosomatidae), and *Trichuris trichiura* (Linnaeus) (Trichinellida: Trichuridae) [38]. However, this technique cannot be utilized to detect intestinal protozoa lacking mitochondria, such as *Blastocystis* spp., *Cryptosporidium* spp. (Tyzzer) (Eucoccidiorida: Cryptosporidiidae), *Entamoeba* spp., and *Giardia* spp. [42]. The DNA barcoding approach is also used to identify arthropod species instead of using morphological identification for which a skilled entomologist is essential. As the complexity and plasticity of arthropods caused the problem of misidentification in morphological identification, DNA barcoding is an alternative or supportive method for species identification. Discriminations using DNA barcoding technique of Culicidae mosquitoes, Psychodidae sandflies, Simuliidae black flies, and organisms in the class Arachnida, such as Buthidae scorpions, Ixodidae ticks, and Trombiculidae mites, have been reported [38,43,44,45].

Apart from its high accuracy, the DNA barcoding technique also showed high sensitivity in parasite and arthropod identifications. Due to the concept of barcode sequence amplification, less specimen volume or an incomplete specimen can be processed by this method. This is because only a small amount of DNA is used as a template for the PCR. As the DNA barcoding technique is based on PCR amplification and DNA sequencing, the diagnoses of many samples can be performed simultaneously, indicating its potential as a high-throughput technology. Moreover, wrong diagnoses by technicians do not occur because the analyses and interpretations are processed via a computer program, except that the wrong nucleotide-sequence data is submitted or the reference sequences in the database are incomplete or missing. Nevertheless, since this technique still requires costly reagents and equipment, it is difficult to use it in resource-constrained clinical settings. Another pitfall of the DNA barcoding technique is the unintentional amplification of nuclear mitochondrial pseudogenes, leading to wrong diagnoses of unknown samples [39,60].

### 3.2. Geometric Morphometric Analysis

Geometric morphometric analysis is a novel approach to parasitological diagnosis. Since it was proposed by Rohlf and Marcus in 1993, this technique has been usually applied to arthropod identification (Figure 1) [61]. Highly skilled parasitologists are not required since interpretations are performed by computer software. Costly reagents and equipment are also not required. The principle of this approach is the statistical analysis of the patterns of shape variation of an anatomical structure among or within groups of samples [62]. A sample is digitized for landmarks on the defined anatomical structure and analyzed for the coordinates of landmarks. This technique was developed from traditional morphometric analysis in which multivariate statistical analysis is applied to sets of morphological variables such as linear length, height, and width [62]. The steps of parasite and arthropod identifications using geometric morphometric analysis are presented in Figure 3. An image of a defined anatomical structure of an unknown sample is recorded using a camera connected to a light or stereo microscope. Landmark digitization is performed on an image of a defined structure using computer software, such as XYOM (https://xyom.io, accessed on 14 August 2021) [63]. The shape-pattern variation is analyzed using statistics, such as principal component analysis, Mahalanobis distance, maximum likelihood, artificial neural network, canonical variates analysis, discriminant function analysis, and related methods. Species identification of an unknown specimen is subsequently processed by comparing the sample coordinates with reference data sets (reference pictures and reference coordinates) or a digital bank of known individuals.

Geometric morphometric analysis is divided into two methods: landmark based and outline based. With the landmark-based method, the differences in the coordinates of the landmarks are analyzed after mathematical removal of non-shape variations. In contrast, the bounding edge (an ordered set of discrete point coordinates) of a defined anatomical structure is compared between specimens in the outline-based method. Nowadays, geometric morphometric analysis is increasingly applied to medical parasite diagnoses. Sumruayphol et al. (2020) utilized geometric morphometric analysis for the morphological identification of the causative agents of fascioliasis, including *F. gigantica*, *F. hepatica*, and *Fasciola* intermediate forms [46]. Adult worms of parasites isolated from the infected cattle’s livers were fixed on the glass slides, soaked in 70% ethanol, and stained with Semichon’s acid-carmine. Pictures of all samples were taken under a stereo microscope and scaled. Five anatomical landmarks were digitized on the right and left sides of the cephalic cone, the right and left sides of the oral sucker, and the end position of the testis by using the CLIC package (www.xyom-clic.eu, accessed on 14 August 2021) [64]. The shape-pattern variation was statistically analyzed using Mahalanobis distance, maximum likelihood, and artificial neural network via XYOM software. The results showed an accuracy of identification of 69%. However, the accuracy increased to 94% when outline-based geometric morphometric analysis was applied, whereas microscopic examination caused an accuracy of identification of 70% [46]. This study highlighted the advantage of geometric morphometric analysis as a new and highly accurate tool for *Fasciola* identification.

García-Sánchez and colleagues (2020) also applied geometric morphometric analysis to the detection of *Trichuris* species eggs. Primate stool samples were collected, smeared onto glass slides, covered with coverslips, and recorded using photographs taken under a light microscope. The pictures were uploaded into a computer program, scaled, and digitized, and the shape patterns were statistically analyzed using principal component analysis, canonical variate analysis, and Mahalanobis distance [65]. In addition, Hugot and Baylac (2007) identified pinworms (Enterobiinae) using a landmark analysis [47]. The shape patterns were analyzed based on 15 landmarks digitized on the caudal bursa of adult pinworms.

In the case of medical arthropod identification, geometric morphometric analysis was broadly applied to many groups of arthropods. For example, classification of the *Aedes* spp. (Meigen) (Diptera: Culicidae) was performed based on an analysis of shape-pattern variation of 20 landmarks digitized on the wing veins [48]. The left wings were detached from the thorax, placed on glass slides, covered with coverslips, and photographed under a stereo microscope. Twenty landmarks were digitized on the wing veins using tps-Dig 1.40 software and analyzed for coordinate variation using principal component analysis. Chiggers (the larval stage of Trombiculidae mites), which is a vector of scrub typhus, was also identified by geometric morphometric analysis. Because of its very small size and as several characteristics are needed for species identification, traditional morphological identification is difficult. Therefore, geometric morphometric analysis was applied to differentiate chiggers using six landmarks on the scutum. This approach showed 100% accuracy for the species identification of the chiggers [49]. Moreover, this technique was used to analyze species of Apidae bees, Formicidae ants, Ixodidae ticks, Psychodidae sandflies, and Reduviidae bugs [50,51,52,53,54].

Compared with conventional parasitological techniques, geometric morphometric analysis is much more accurate, more user friendly, less time consuming, and less labor intensive. Unlike DNA barcoding analysis, it is also a low-cost technique. Neither expensive reagents nor sophisticated technology is needed. Only specific software is required to digitize the landmarks and statistically analyze the coordinate variations. Apart from its usefulness for parasite and arthropod identifications, this advanced technique also draws upon studies of phenotypic variations, such as heritable consequences, genetic assimilation, epigenetics, and hybridism caused by environmental impact within a species [66]. Furthermore, it can be applied as an entomological surveillance tool for vector control [66]. However, unlike the DNA barcoding technique, an incomplete specimen cannot be diagnosed by this method since the analysis is based on a comparison of anatomical structures. Since many picture samples can be analyzed simultaneously, geometric morphometric analysis is an advanced technique that can be utilized as a high-throughput technology for parasite and arthropod identifications as well as a tool to study variations within a species.

### 3.3. Artificial Intelligence Technology

Artificial intelligence (AI) is an advanced computer science technology that emerged in 1956 (Figure 1) [67]. Since its conception, this technology has been applied to multiple sectors. However, its application to the field of parasitological diagnosis only occurred recently. It is an innovation that aims to simulate human intelligence, such as recognition and problem-solving skills, in machines and computer programs. It is governed by a working of a set of rules called an “algorithm”, which is trained via a machine-learning process [68]. Currently, AI algorithms are highly developed, and deep-learning algorithms have emerged. These include convolutional neural networks (CNNs) that have been extensively applied to medicine, especially for medical image processing. The main steps of AI comprise data preparation, data entry, data processing, data learning, modeling, and testing. Because the accuracy of AI processing depends on machine learning, the quality, quantity, and variations of the training materials are critical factors.

AI is presently utilized in several broad fields. For instance, AI has been applied as a diagnostic tool for infectious diseases in pathological, microbiological, and parasitological laboratories. As to microbiological laboratories, Egli et al. (2020) reported the use of AI in pre- to post-analytical processes [69]. In the pre-analytical processes, AI was applied to the chatbot to guide specimen collection, transportation, diagnostic approach, unit cost, and evaluation of turnaround time. Automated weighting was also utilized as a tool to assess specimen volume and provide feedback. In the case of the analytical processes, automated microscopy was used to capture blood smears and classify bacteria based on Gram staining [70]. An automated plate-reading system was utilized to detect bacterial growth on an agar plate of a urine culture with 97.1% sensitivity and 93.6% specificity and to identify bacterial species based on colony morphology on chromogenic agar with 99.7% accuracy [71]. Moreover, an advanced expert system was applied to interpret drug-resistance profiles [72]. Finally, dashboards were used in the post-analytical process to summarize analytical results to facilitate the determination of patient intervention strategies by physicians.

For parasitological diagnoses, AI has recently been applied to malaria diagnoses. Blood samples were smeared onto glass slides, stained with Giemsa, and observed and recorded by microscopy. All images were transferred into and preprocessed by computer programs, including noise removal, contrast improvement, illumination, and staining correction. Red blood cells and other objects were segmented based on shape, size, texture, and color by different algorithms, such as edge detection, watershed, and neural network algorithms. All features presented on an image were then extracted and selected. The mathematical feature extraction was based on the characteristics of the features (color, texture, and typical appearance), and the selection involved principal component analysis, F-statistic, one-way analysis of variance, and the like. For example, infected red blood cells were extracted from uninfected red blood cells for parasitemia calculation. Finally, identification of *Plasmodium* species was processed based on special characteristics, such as ring- or band-form trophozoites, using deep neural network or support vector machine algorithms (Figure 4) [55]. Torres et al. (2018) reported that automated microscopy can be used as a routine diagnostic tool for malaria instead of human technicians with a 72.0% sensitivity and an 85.0% specificity [56]. Compared with microscopic examination by technicians, autoscoping with a convolutional neural network algorithm can detect *Plasmodium* spp. from 700 samples of Giemsa-stained blood smears with the same accuracy.

Furthermore, AI was applied to detect the ova of soil-transmitted helminths and trematodes. Holmström et al. (2017) used a mobile digital microscope with a deep-learning-based, computer-vision algorithm to detect the ova of *A. lumbricoides*, *T. trichiura*, hookworms, and *S. haematobium* from stool and urine samples, achieving an 83.3–100.0% sensitivity [57]. The stool specimens were subsequently concentrated with a formalin ethyl acetate concentration and fixed with acrylamide solution. Twenty microliters of the mixtures were dropped onto glass slides, covered with coverslips, and coated with mounting media. Pictures were then scanned using a small, lightweight, inexpensive, and cloud-connected digital microscope. The images were processed, and ova were identified using deep-learning-based, machine-learning algorithms. Mathison et al. (2020) applied AI to the diagnosis of medical protozoa, such as *Blastocystis* spp., *Chilomastix mesnili* (Wenyon) (Retortamonadida: Retortamonadidae), *Dientamoeba fragilis* (Jeeps et Dobell) (Tritrichomonadida: Dientamoebidae), *Endolimax nana* (Wenyon and O’Connor) (Mastigamoebida: Entamoebidae), *Entamoeba* spp., and *G. duodenalis* [58]. Stool samples smeared onto glass slides were stained with trichrome; pictures were scanned using digital slide scanning; and protozoa were detected with a computer program using a deep convolutional neural network. The intestinal protozoa were detected with a 98.8% positive agreement with the findings of manual microscopy by trained personnel. Altogether, these studies highlight that medical parasites can be diagnosed with AI technology, inexpensive equipment, and non-specialists. Moreover, AI technology can be used as a high-throughput diagnostic tool for mass screening and for point-of-care testing, making it suitable for low-resource healthcare settings.

As to medical arthropods, deep-learning approaches were also utilized for species identification of mosquito vectors. The deep-learning models were based on the “you-only-look-once” (YOLO) algorithm. They were used to identify *Aedes* species, such as *A. aegypti* and *A. albopictus*; *Culex* species (Linnaeus) (Diptera: Culicidae), like *C. gelidus*, *C. quinquefasciatus,* and *C. vishnui*; and *Mansonia* species (Blanchard) (Diptera: Culicidae), for example, *M. annulifera*, *M. indiana*, and *M. uniformis*. Moreover, *Armigeres* spp. (Coquillett) (Diptera: Culicidae), *Anopheles* spp., *Musca domestica* (Linnaeus) (Diptera: Muscidae), *Trigona apicalis* (Jurine) (Hymenoptera: Apidae), and *Oryzaephilus surinamensis* (Ganglbauer) (Coleoptera: Silvanidae) were also identified by this method with a 99.0% precision and 92.4% sensitivity [59].

These data indicate that AI has gradually been transformed for use in practical applications for parasite and arthropod identifications. The benefits of this advanced technology are not only its high accuracy, sensitivity, and specificity but also its short processing time and relatively low need for human resources. Significantly, this technology does not require the use of technical staff who are highly skilled in parasitology, and it shows great potential for use as a high-throughput diagnostic tool.

## 4. Future Perspectives and Challenges of Parasite and Arthropod Diagnostics

In the near future, both digitalization and automated machines and processes are expected to have a critical impact on the way that parasitological laboratories work. In the case of digitalization, in which all information is converted into a digital form that can be processed by a computer, advanced technologies, such as DNA barcoding, geometric morphometrics, and AI, are very appealing. All laboratory results are converted into a digital form, analyzed, and interpreted by computer programs. Only sample preparation, machine operation, and computer processing are controlled by human technicians with minimal training. Therefore, interpretation bias is limited. Nowadays, mobile-phone-based microscopes are receiving increasing attention since they are lightweight and easily carried. These tools can digitize, save, and transfer images to cloud systems for further analysis. They have been used to detect *Schistosoma* spp. and intestinal protozoa [73]. Recently, smartphone-based optofluidic lab-on-a-chip was developed to detect malaria from blood [74]. Moreover, metagenomic analysis is a new promising tool to identify genus and species of parasites, such as *Cryptosporidium* spp., *Plasmodium* spp., and *Toxoplasma* spp. (Nicolle and Manceaux) (Eucoccidiorida: Sarcocystidae) [75,76,77]. This technique increases specificity and accuracy of organism detection because it increases the number of markers for characterization in a species level. Nowadays, this approach is low cost and shows suitability for in-field and clinical uses due to application of Oxford Nanopore Technologies (ONT) [78]. For automatic pre- and post-analytical processes, they are preferred by many clinical laboratories because they can decrease wrong diagnoses, processing times, and labor requirements. Such tools are anticipated to have an enormous impact on the daily routines of laboratory staff. All parasitological staff will need to be prepared to adapt to the emerging technological advances in parasitology. However, there are several challenges in terms of technology and resources in the present day. In addition, the interpretation processes of all of above approaches require reference data sets for comparison purposes. Therefore, challenges lie ahead in ensuring that there is coverage of all medical parasites and arthropods in the databases that are deposited in computer software. Nowadays, despite several new approaches having emerged, detection capabilities are still limited to a narrow range of species. Furthermore, there is a pressing need to develop user-friendly and point-of-care parasite and arthropod testing that is suitable for use in both high- and low-resource healthcare settings [79,80].

## 5. Conclusions

Although traditional methods, like microscopic and immunological examination, are still considered the gold standard for the diagnosis of parasitic and arthropod diseases, limitations in terms of their sensitivity, specificity, and wrong diagnoses by human technicians are recognized as major problems. Even though various molecular techniques have been developed to overcome these pitfalls, wrong diagnoses remain the main problem because interpretations are still made by human diagnosticians. The high cost of reagents and equipment is also a limiting factor for the use of molecular techniques. Therefore, alternative diagnostic tools for parasite and arthropod identifications are being developed; they employ computer program-based methods without the need for costly reagents and equipment. Geometric morphometric analysis and AI technology are novel approaches that appear to be ideal diagnostic tools. They have demonstrated high sensitivities, specificities, and accuracies, and they have the potential to be used as high-throughput technologies for the diagnosis of parasitic and arthropod diseases. Moreover, they do not require the use of costly reagents and equipment; nor are highly skilled technicians needed. Since it is recognized that these more sensitive and accurate methods could control the mortality of parasitic diseases, it can be stated that these novel approaches are very promising tools for parasite and arthropod diagnoses. However, the limitations of using these computer-assisted techniques in real clinical settings still occur, such as species coverage and internet access. Hence, combining conventional and advanced methods may decrease limitations and improved diagnosis precision using the advantages of each technique.

## Figures and Tables

**Figure 1 diagnostics-11-01545-f001:**
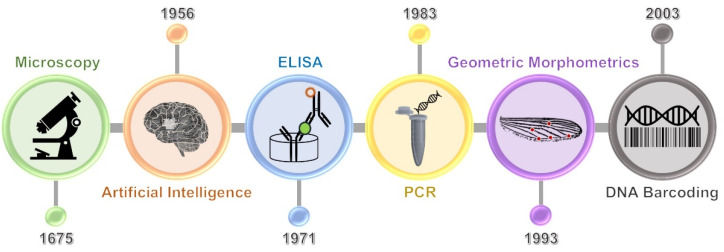
Timeline of technologies for medical parasite and arthropod diagnoses. Microscopy has been applied for pathogen and insect identifications for several centuries. To improve detection efficacy, a range of techniques have emerged. They encompass artificial intelligence, ELISA, PCR, geometric morphometrics, and DNA barcoding techniques.

**Figure 2 diagnostics-11-01545-f002:**
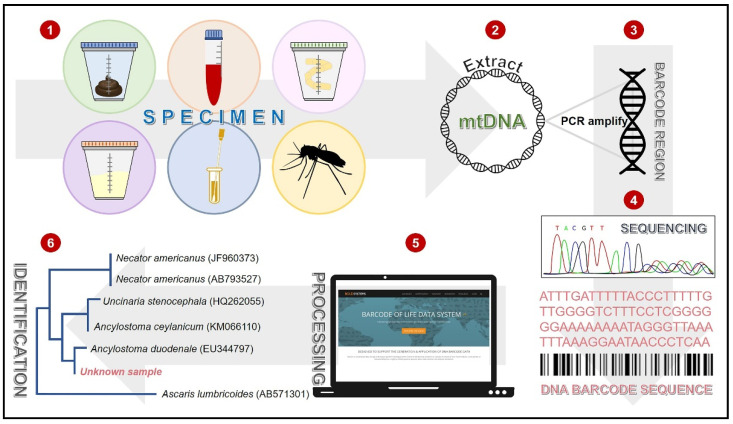
Steps of DNA barcoding technique for medical parasite and arthropod diagnoses. DNA is extracted from a clinical specimen (**1**), and a barcode region is amplified using the universal primers (**2**,**3**). The PCR product is purified and submitted for nucleotide sequencing (**4**). This sequence is subsequently analyzed for species identification based on the nucleotide sequence similarity to the referenced sequences in the BOLD database (**5**,**6**).

**Figure 3 diagnostics-11-01545-f003:**
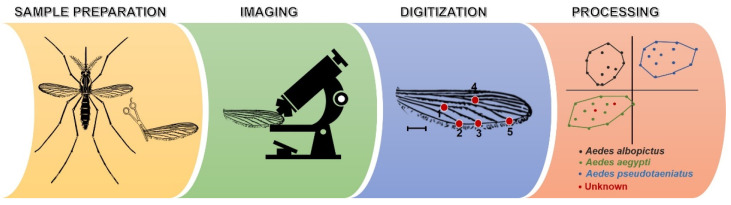
Steps of medical parasite and arthropod diagnoses using geometric morphometric analysis. An unknown sample is prepared on a glass slide or any solid support. A defined structure of an unknown sample is photographed and submitted to XYOM software (https://xyom.io, accessed on 14 August 2021) [63] or related tools. The image file input is scaled, landmark digitized, and statistically analyzed.

**Figure 4 diagnostics-11-01545-f004:**
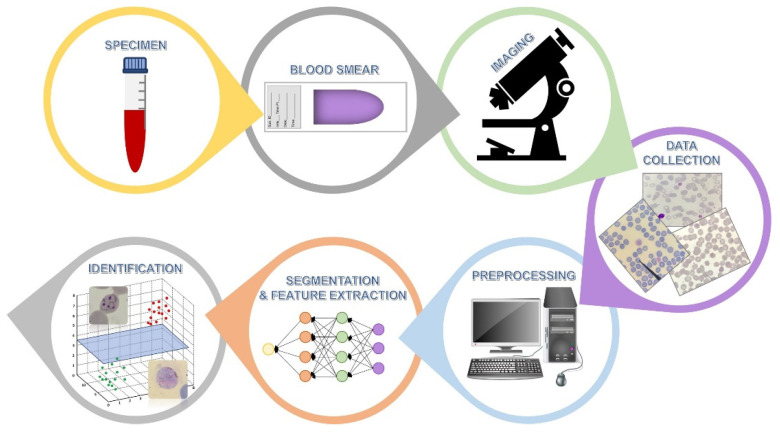
Malaria diagnosis by artificial intelligence technology. After performing a blood smear and staining, image acquisition, preprocessing, cell segmentation, feature extraction and selection, and parasite identification were performed. Several mathematic models and algorithms were applied to identify the *Plasmodium* species.

**Table 2 diagnostics-11-01545-t002:** Recent advanced tools for diagnosis of medical parasites and arthropods.

Methods	Principle	References
DNA barcoding technique	Analysis of barcode sequence	[38,39,43,44,45]
Geometric morphometric analysis	Statistical analysis of shape pattern variation of an anatomical structure	[46,47,48,49,50,51,52,53,54]
Artificial intelligence technology	Analysis of picture using the trained algorithms	[55,56,57,58,59]

## Data Availability

Not applicable.

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
