# Peer review of "State-of-the-Art Techniques for Diagnosis of Medical Parasites and Arthropods"

_diagnostics, 2021, doi:10.3390/diagnostics11091545_

Round 1
Reviewer 1 Report
The review entitled " State-of-the-Art Techniques for Diagnosis of Medical Parasites and Arthropods" summarizes current diagnosis techniques applied in medical studies with a major focus on computational approaches that can help increase diagnosis precision. Overall the manuscript was well written and covers almost everything proposed by the authors.
Here are some comments:
- Table1 seems to cite the wrong citation for all methodologies presented. (I believe that it was reference 12); I also would recommend adding the original reference for each method instead of the book.
- line-94-95: maybe add why PCR and antigen detection are still not used for malaria. Usually, tropical and neglected diseases are highly endemic in regions with low resources, and Giemsa and microscopic examination are still the cheapest method to deal with the high demand for tests.
- line-150: add "or equipment other than computers")
- I missed citations in some paragraphs, for example on lines 160-167 the whole paragraph does not have any citation. I can see at least three affirmative that need a proper citation (most of them already cited in other places in the review). Please add those and avoid paragraphs without citation in a review.
- line 182- Indeed BOLD is a good database, but I would also add about the avaiablity of genomic information available in larger databases such as GenBank. Today many tools used to detect genus or species from metagenomic analysis build their database from Genbank. This helps the increase of specificity and accuracy of organism detection and increase the number of markers for characterization in a species level.
- I believe that the group did a great job showing the DNA barcoding, geometric morphometric analysis, and artificial inteligence as the new advanced approaches for diagnoses. But currently, there are also whole genome and aplicon sequencing approaches with low cost using nanopore technologies that can evaluate the not just one marker such as normal barcodes (ITS) but a broad number of markers in a metagenomic level. I would recomend the authors to add a section about metagenomic applications in diagnosis of parasitic diseaes or at least add about it in section 4 about future perspectives. For example ONT has flongle cells that cost in average $100 and can sequence up to 3GB bases. This means that several markers can be sequenced and give a more accurate overview of the disease (Cryptosporidium, Toxoplasma, and Plasmodium, for example, have the entire genome size s averaging from 9Mb-60Mb). Of course for bigger genomes such as arthropods, these metagenomes would be expansive.
Author Response
Thank you so much for your generous and brilliant comments. Please see the attachment for the point-by-point revision.

Reviewer 2 Report
The manuscript “State-of-the-art techniques for diagnosis of medical parasites and arthropods” by P. Ruenchit (diagnostics-1334895), although dealing with an interesting subject, is not suitable for publication in its present form and requires a major and complete revision. Unless this complete revision is performed, the manuscript can not be reconsidered.
First, all reference numbers cited in the text of the manuscript are wrong. These errors made very difficult the task of reviewing it. There is an initial error in the References section (two references enlisted as n. 1) but there are also citations either wrong or missing in the text. The author should carefully check all the manuscript (including Tables) for the correct reference numbers and add more references when required.
General comments
A major flaw of the manuscript is the fact that the author used other reviews for his/her review without considering the original works on which the reviews were based. Thus, his/her citations appeared too general and often unrelated to the text.
Another general remark is the excessive trust placed by the author in computer-assisted techniques (see lines 63-71, 148-151 and 215-217). All techniques have their advantages and disadvantages and the approach to correct identification of parasites of medical interest should be an integrated one, using the advantages of all techniques that could be employed.
The text of the manuscript should be completely revised and shortened: it is exceedingly verbose in many parts (see for example the unnecessarily detailed discussion of works about geometric morphometric analysis and artificial intelligence), and there are repetitions of the same concepts in several paragraphs and sections of the manuscript that should be amended (see detailed comments).
The text of the entire manuscript should also be revised for English style, using simpler and clearer sentences, and clarifying the meaning of some of them (see detailed comments).
Detailed comments
Lines 32-36. The phyla should be indicated in phylogenetic order: first “Protozoa” (that should be replaced by the correct systematic name “Protista”), then Trematoda, Cestoda and Nematoda. In addition, the species should be enlisted in alphabetical order. When a species is cited for the first time, it should be indicated on the base of international taxonomical rules, indicating the authority, the order and the family, as follows:
Wuchereria bancrofti (Cobbold) (Spirurida: Onchocercidae)
Further citations of the same species should be W. bancrofti.
Lines 40-41. Is the author sure of his/her statement? Parasitic diseases are widespread in the world: does the author refer to Thailand? This sentence should be clarified.
Lines 44-48. For clarity, arthropods should be indicated with correct systematic criteria, grouping together the insects and the other arthropods.
Line 49. There are many other species of Diptera causing cutaneous myiasis and other types of obligatory and facultative myiasis, not only Chrysomya megalocephala. For example, there are Cochliomyia hominivorax, Wohlfahrtia magnifica, Lucilia cuprina, L. sericata, L. caesar and so on. Moreover, there are also species causing furuncular myiasis, such as Dermatobia hominis and Cordylobia anthropophaga.
Lines 63-71, 148-151 and 215-216. Is the author sure that “errors from human diagnosticians” (that should be replaced by “errors committed by physicians”) and “misdiagnoses” (that should be replaced by “wrong diagnoses”) could be avoided only by using computer programs? Computers are operated by humans, and inserting the wrong data in computers may as well cause errors. See line 145: the author states that “skilled technicians” are required to interpret results obtained from molecular techniques.
Lines 83-87. These sentences state the obvious: the part from line 85 to line 87 should be deleted.
Lines 97-98. There are specific university courses preparing experts with “a high degree of expertise for interpretation”.
Lines 194-198. Again, the author states that a “skilled entomologist” is required for “morpho-taxonomy” (that should be replaced by “morphological identification”). The same ability (by a molecular biologist or a geneticist) is obviously required to interpret DNA barcoding data.
Lines 198-200. Families, subfamilies, classes and common names are mixed together: please uniform the list based on correct classification criteria.
Line 212. What do the author mean by “a small number of nucleic acids”? Perhaps “a small amount of DNA”? Please clarify the sentence.
Line 215. Concerning DNA barcoding, as mentioned before it is not enough to insert data in the computer program to obtain a result, especially when the reference sequences in the database are incomplete or missing. Again, it is necessary a “skilled technician” (or, more correctly, an expert in the field) for a correct interpretation of the data.
Line 254. Delete “being”.
Line 262. No reference is provided about the CLIC package.
Lines 264-266. Why does the author compare the data in terms as “accuracy of identification” for statistical analysis and geometric morphometric analysis, and in terms of “inaccuracy” for microscopic examination, suggesting a “bias” for this technique? The author should replace the sentence in line 266 as “an accuracy of identification of 70%”. In addition, no reference is provided for these data.
Figures
In Figure 2, it should be specified that the image of circular DNA refers to mitochondrial DNA, since animal (thus eukaryotic) DNA is linear. Replace “DNA” in the figure with “mitochondrial DNA” or “mtDNA”. In addition, in the same figure the sequence of DNA barcoding steps is unclear: please indicate each step with a letter.
Tables
In Table 1 (Conventional methods for diagnosis…) the second line (“Gross” and “Gross examination”) should not be in bold. Also, what do the author mean by “Diagnostic Tool of Parasite and Arthropod”? The difference should be clearly explained in the table caption.
Moreover, column 3 (“Parasite”) reports the same word (“Yes”) for all lines except “Skin scraping technique”: this column is superfluous and the information should not be enlisted in the table but explained in the text.
All the above comments are also true for Table 2: column 2 and 3 should be deleted and explained in the text.
Author Response

(The authors gave the same response as above.)

Reviewer 3 Report
This review summarises the state-of-the-art techniques for diagnosis of medical parasites with an emphasis on the arthropods. It is a very well written article focused not only on the advances of the techniques for parasite identification, but also provides an overview on their drawbacks.
I have found just two typos when reading the MS:
- 103: hookwormS, i.e. it shuld be in plural
- 164: either “take” or “occur”, but not both
Recommendation
I would rather suggest reconsidering the title and omitting “and Arthropods” as the parasitic arthropods and part of the parasites with medical importance.
Author Response

(The authors gave the same response as above.)

Round 2
Reviewer 2 Report
The manuscript “State-of-the-Art Techniques for Diagnosis of Medical Parasites and Arthropods” (ID 1334895) by P. Ruenchit has been considerably improved in its revised version, but there are a few minor corrections that should be made before it is accepted for publication.
First, many systematic names were cited along the manuscript, and these additions were appreciated because they improved the scientific value of the manuscript. However, all nomenclatures of species and genera (including authorities) require a final careful check because there are still some errors, for example:
Line 53. “Cimex spp. (Hemiptera: Cimicidae)” should be “Cimex spp. (Rhynchota: Cimicidae)”.
Lines 410-413. After checking the reference cited by the author, the nomenclature should be corrected as follows: “Armigeres spp. (Coquillett) (Diptera: Culicidae), Anopheles spp., Musca domestica (Linnaeus) (Diptera: Muscidae), Trigona apicalis (Jurine) (Hymenoptera: Apidae), and Oryzaephilus surinamensis (Ganglbauer) (Coleoptera: Silvanidae)”.
Other corrections that should be made to the manuscript are:
Line 52. Does the list of “medical arthropods” refer to Thailand or generally to the world? It should be specified.
Lines 108-110. The sentences “It is because of the cost of microscopic examination is very low compared to PCR and antigen detection. So, it can serve the high demand for tests in the endemic areas with low resources” are unclear and should be clarified. Perhaps the author meant: “This is due to the very low cost of microscopic examination compared to PCR and antigen detection, meeting the high demand for tests in endemic areas with low resources”.
Lines 224-227. Figure 2. The sequence of DNA barcoding steps was defined by numeric signs as suggested, but the numbers should be added to the figure caption, as follows:
“Figure 2. Steps of DNA barcoding technique for medical parasite and arthropod diagnoses. DNA is extracted from a clinical specimen (1) and a barcode region is amplified using the universal primers (2, 3). The PCR product is purified and submitted for nucleotide sequencing (4). This sequence is subsequently analyzed for species identification based on the nucleotide sequence similarity to the referenced sequences in the BOLD database (5, 6).”
Line 392. “Mathison and colleagues (2020)” should be “Mathison et al. (2020)”.
Line 434. Please delete “an advanced DNA barcoding technique”.
Author Response
Thank you again for your kind suggestions. I have learned a lot from you. Please see the attachment for my revision.
